# Room temperature organic magnets derived from $sp^3$ functionalized graphene

Jiří Tuček[1], Kateřina Holá[1], Athanasios B. Bourlinos[1,2], Piotr Błoński[1], Aristides Bakandritsos[1], Juri Ugolotti[1], Matúš Dubecký[1], František Karlický[1], Václav Ranc[1], Klára Čépe[1], Michal Otyepka[1] & Radek Zbořil[1]

Materials based on metallic elements that have $d$ orbitals and exhibit room temperature magnetism have been known for centuries and applied in a huge range of technologies. Development of room temperature carbon magnets containing exclusively $sp$ orbitals is viewed as great challenge in chemistry, physics, spintronics and materials science. Here we describe a series of room temperature organic magnets prepared by a simple and controllable route based on the substitution of fluorine atoms in fluorographene with hydroxyl groups. Depending on the chemical composition (an F/OH ratio) and $sp^3$ coverage, these new graphene derivatives show room temperature antiferromagnetic ordering, which has never been observed for any $sp$-based materials. Such 2D magnets undergo a transition to a ferromagnetic state at low temperatures, showing an extraordinarily high magnetic moment. The developed theoretical model addresses the origin of the room temperature magnetism in terms of $sp^2$-conjugated diradical motifs embedded in an $sp^3$ matrix and superexchange interactions via –OH functionalization.

[1] Regional Centre of Advanced Technologies and Materials, Department of Physical Chemistry, Faculty of Science, Palacky University in Olomouc, Slechtitelu 27, Olomouc 783 71, Czech Republic. [2] Physics Department, University of Ioannina, Ioannina 45110, Greece. Correspondence and requests for materials should be addressed to R.Z. (email: radek.zboril@upol.cz).

Since the first isolation of graphene—2D carbon allotrope—in 2004 (ref. 1), vast efforts have been made to understand its unique mechanical, electronic, optical and transport properties[2–8]. Among other properties, it exhibits superior mechanical strength,[4] a very large specific surface area[9], high carrier mobility[2], transparency[10] and thermal conductivity[6]. Moreover, several peculiar physical phenomena have been observed in graphene such as the ambipolar effect[1], room temperature half-integer quantum Hall effect[7], nonlinear Kerr effect[11] and Casimir effect[12]. Because of its remarkable properties, graphene has great potential in a broad portfolio of technical applications[13] including robust lightweight, thin and flexible display screens[14], field-effect and ballistic transistors[15], spin transistors and spin logic devices[16], photosensitive transistors[17], organic photovoltaic cells[18], organic light-emitting diodes[19] and conductive plates in supercapacitors and lithium-ion, lithium-sulfur and lithium-air batteries[9,20,21].

However, the practical applications of graphene are limited by its zero band gap, hydrophobicity and absence of long-range magnetic ordering. One potentially effective way of eliminating these drawbacks is to instead use covalently functionalized graphene derivatives in which specific atoms or functional groups are covalently bound to the graphene sheet to tune its physicochemical and biochemical properties[22]. Important examples of covalently modified graphene derivatives include graphene oxide[23], graphane[24] and fluorographene[25]. In particular, graphene oxide and fluorographene can be further readily functionalized with various other functional groups (for example, –Cl, –I and –SH), offering further scope for band gap tuning and the introduction of new electronic, optical and sensing properties[23,26,27].

Covalent functionalization has thus made it possible to produce graphene derivatives with modified band gap properties and altered hydrophilicity/hydrophobicity[22,28]. However, development of graphene derivative with room temperature magnetic behaviour is a major unaddressed challenge despite the investment of considerable effort into imprinting stable magnetic centres into graphitic structures and inducing long-range magnetic ordering across 2D carbon networks[29,30]. Several key strategies have been suggested to induce spin-carrying $sp^3$ (paramagnetic) states in graphene-related structures[29,30] such as formation of defects and vacancies,[30] insertion of non-carbon atoms (for example, boron, nitrogen and sulfur) into the graphene lattice[31–33], cutting of graphene sheets creating edges with a specific geometry (for example, zigzag graphene nanoribbons)[34], covalent functionalization with functional groups[29,30,35–37], light-atom adsorption (that is, adatoms)[38–40], transition-metal-atom adsorption[41] and electric field engineering[42]. Particularly, the pioneering work of Nair et al.[43] suggested that it may be possible to imprint paramagnetic centres into graphene by combining defects arising from partial fluorination with appropriate (C–F) functionalization. All these approaches demonstrate the possibility to combine various sources for creation and $\pi$-electron system-based coupling of spin-carrying $sp^3$ states to induce magnetism into graphene. However, $\pi$-electron system-mediated interactions are weak and, hence, magnetic ordering collapses at relatively low temperatures ($< 100$ K). Thus, room temperature magnetism in graphene and graphene derivatives, maintained by the $\pi$-electron system, is heavily questioned in literature[13]; if such a behaviour was experimentally observed, the presence of impurities of transition metal origin (Fe, Ni, Co), originating from either the synthesis itself or sample handling, was not properly excluded, leading to misinterpretation of results and inaccurate conclusions[13].

Here we report a discovery of organic graphene-based magnets with a magnetic ordering sustainable up to room temperature due to the suitable $sp^3$ functionalization. A series of magnetic carbons designated as hydroxofluorographenes are prepared from fluorographene by exchanging some of its fluorine atoms for hydroxyl groups. The chemical composition of hydroxofluorographenes can be controlled through reaction conditions and choice of –OH-containing precursors. Strikingly, hydroxofluorographenes with an appropriate composition (an F/OH ratio) exhibit antiferromagnetic ordering at room temperature, a magnetic behaviour not previously observed for any graphene derivative or $sp$-based material. At low temperatures, these hydroxofluorographenes undergo a transition to a ferromagnetic state with one of the highest magnetization values reported among graphene-based material. Based on the set of experimental data and high throughput first principles calculations on a large number of atomic configurations with varying F/OH ratio, we establish that the unique magnetism is attributed to a network of functionalization-induced $sp^2$-conjugated carbon diradical motifs embedded in an $sp^3$ matrix, and the ability of –OH groups to stabilize magnetically ordered state up to room temperature due to emergence of superexchange interactions. Moreover, the suggested theoretical model for hydroxofluorographene system has a universal nature and covers both 'diradical motifs-induced magnetism' appearing at high $sp^3$ coverages and sustaining up to room temperature and 'defect-induced magnetism', which emerges at lower degrees of $sp^3$ functionalization with limited sustainability at higher temperatures.

## Results

**Physicochemical properties of hydroxofluorographenes.** The synthesis of hydroxofluorographenes is based on the chemistry of fluorographene (GF), which has been investigated experimentally and computationally by our group, leading to the development of practical procedures for the direct nucleophilic substitution of fluorine[26,27]. The series of five hydroxofluorographenes differing in degree of $sp^3$ functionalization, an F/OH ratio, and magnetic features were prepared by ultrasonic exfoliation of fluorographite ($C_1F_1$) in N,N-dimethylformamide (DMF), after which the exfoliated material was treated with –OH-containing precursors for various reaction temperatures and time. For detailed chemical, structural, morphological and magnetic characterization, we have selected the representative sample, denoted as G(OH)F, prepared by reaction of fluorographene with tetramethylammonium hydroxide in DMF for 3 days (Fig. 1; for more details, see 'Methods' section below).

The overall chemical composition of G(OH)F sample was determined by X-ray photoelectron spectroscopy (XPS) analysis (Supplementary Fig. 1), revealing the average contents of oxygen, fluorine, carbon and nitrogen to be 6.1, 27.2, 65.4 and 1.3 at.%, respectively. To exclude the presence of metals, which would affect magnetic features of the system, we performed inductively coupled plasma mass spectrometry (ICP-MS) analysis confirming their negligible contents generally below 24 ppm (Supplementary Table 1). The C/F ratio of G(OH)F (2.4/1) found from XPS analysis is greater than that of its precursor, fluorographene (1/1), reflecting the sample's partial defluorination and formation of some aromatic $sp^2$ regions during G(OH)F synthesis. The presence of oxygen, fluorine, carbon and nitrogen was inferred from energy-dispersive X-ray (EDX) spectrum measured for the G(OH)F sample (Supplementary Fig. 2). The small content of nitrogen comes from DMF solvent residues as clearly seen in the thermogravimetric curve (Supplementary Fig. 3a) exhibiting a negligible loss (0.41 wt.%) up to 100 °C due to release of adsorbed water, while the secondary weight decrease (1.11 wt.%) between 108 and 140 °C is evidently related to DMF evolution. To confirm the presence of fluorine and hydroxyl groups in the G(OH)F structure, the G(OH)F sample was

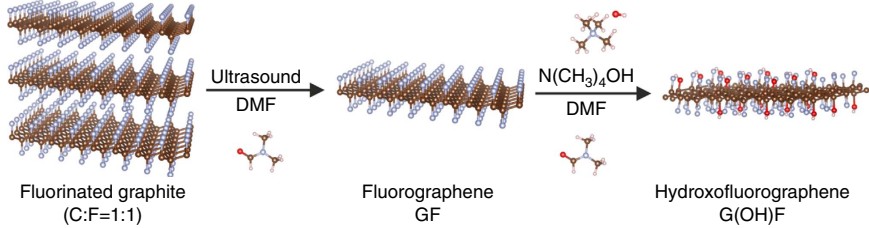

**Figure 1 | Representative preparation of G(OH)F.** The scheme depicting the chemical procedure towards G(OH)F.

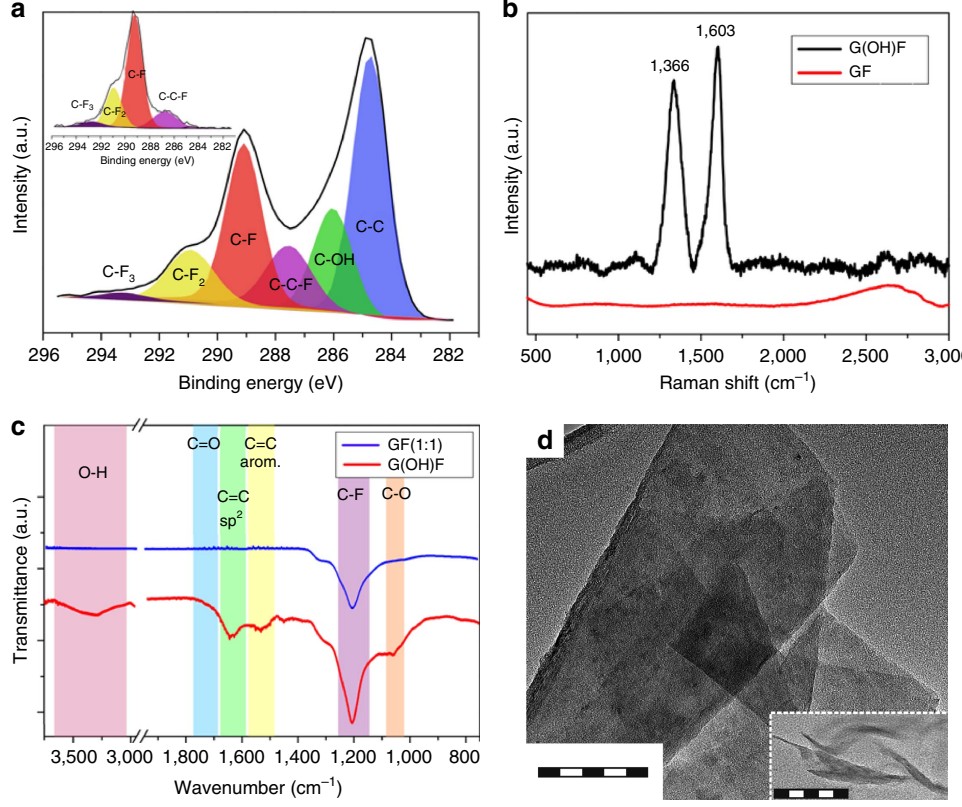

**Figure 2 | Physicochemical characterization of GF and G(OH)F.** (**a**) High-resolution C 1s spectrum of G(OH)F and GF precursor (inset). (**b**) Raman spectra of G(OH)F (black line) and GF (red line). (**c**) FT-IR spectra of G(OH)F (red line) and GF (blue line). (**d**) HRTEM image (scale bar, 100 nm) of a G(OH)F sheet with an inset (scale bar, 100 nm) demonstrating a single-sheet character of G(OH)F.

subjected to thermal decomposition under nitrogen atmosphere, heating from room temperature to 900 °C, and the release of gases and ions was monitored by mass spectrometry (Supplementary Fig. 3b). The release of hydroxyl ions started once the temperature rose above 200 °C; this was attributed to the removal of covalently attached –OH groups because the adsorbed water molecules were released at temperatures of up to 100 °C. The two peaks appearing in the evolved gas profile for the –OH groups can be well explained in terms of the stability of –OH groups in G(OH)F strongly dependent on their local environment (Supplementary Fig. 3c). Hence, the first peak, with maximum at 220 °C, corresponds to the evolution of less stable –OH groups, which are released before any evolution of fluorine, while the second peak, centred at 260 °C, corresponds to –OH groups released after the onset of fluorine evolution, which would radically change their local environment and stability (Supplementary Fig. 3c). The fluorine ions were detected in a broad range of temperatures with the maximum of fluorine released at 500 °C. Importantly, carbon dioxide and

carbon monoxide, whose release typically indicates the presence of carboxylic and epoxy groups, were not detected at any temperature.

The successful attachment of hydroxyl groups was also confirmed by high-resolution C 1s XPS data (Fig. 2a) showing that the product contained C–O and C–C bonds, which were not present in the XPS pattern of the GF precursor (inset in Fig. 2a). Formation of aromatic $sp^2$ regions was also indicated by Raman spectroscopy (Fig. 2b): Raman spectrum of the G(OH)F sample exhibited a characteristic G-band at 1,603 cm$^{-1}$ and a disorder-induced D-band at 1,366 cm$^{-1}$, whereas pristine fluorographene is Raman-inactive[44]. The Fourier transform infrared (FT-IR) spectrum of G(OH)F featured vibrations in the range of 1,510–1,670 cm$^{-1}$ corresponding to aromatic regions, and C–F vibrations at 1,200 cm$^{-1}$ (Fig. 2c). Importantly, its FT-IR spectrum also showed peaks associated with covalent C–OH bonds that were not observed in the FT-IR spectrum of fluorographene, namely a C–O vibration peak at 1,058 cm$^{-1}$ and a broad O–H vibration peak at

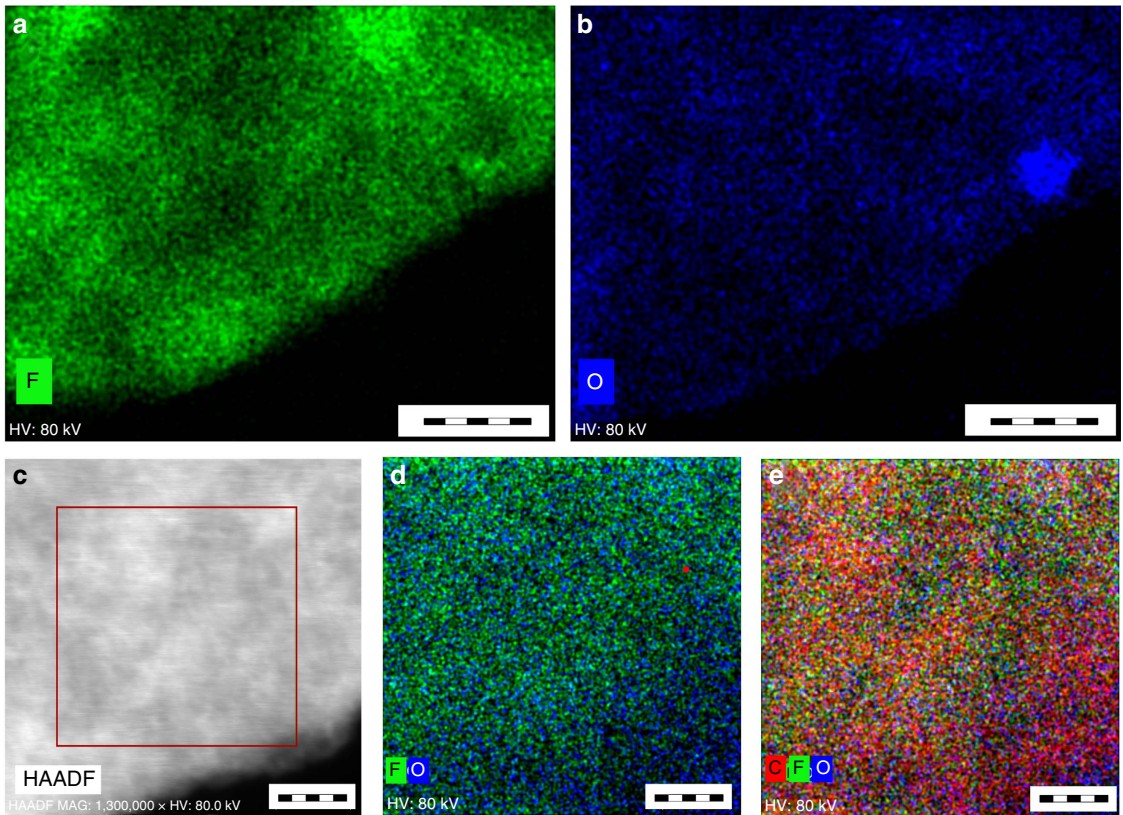

**Figure 3 | Chemical mapping of a G(OH)F sheet by STEM–HAADF.** (**a**) Distribution of fluorine atoms on the G(OH)F sheet (scale bar, 50 nm). (**b**) Distribution of oxygen atoms on the G(OH)F sheet (scale bar, 50 nm). (**c**) STEM/HAADF image of the G(OH)F sheet (scale bar, 10 nm). (**d**) Combined F/O chemical mapping of the G(OH)F sheet (scale bar, 6 nm). (**e**) Combined C/F/O chemical mapping of the G(OH)F sheet (scale bar, 6 nm).

3,250 cm$^{-1}$. These results indicate that a well-defined covalently modified graphene derivative containing aromatic rings and two characteristic functional groups, C–OH and C–F, can be prepared from fluorographene via a simple synthetic procedure involving chemical exfoliation followed by nucleophilic substitution. The stoichiometric formula for this representative material is approximately $C_{18}(OH)_{1.8}F_{7.2}$ with $C_{18}$ as a supercell used in the theoretical model (see below).

Atomic force microscopy experiments indicated that after the exfoliation process, the sample consisted of single-layered sheets less than 1 nm thick (Supplementary Fig. 4), however, a few-layered sheets with thickness of several nanometres were observed as well. Similarly, very thin highly transparent single sheets were observed by high-resolution transmission electron microscopy (HRTEM) (Fig. 2d). The distribution of fluorine and oxygen within the sheet was investigated by scanning transmission electron microscope–high-angle annular dark-field imaging (STEM–HAADF) chemical mapping (Fig. 3), which showed that fluorine and oxygen (–OH) groups were distributed rather homogeneously within the G(OH)F structure, without any large fluorine (or oxygen) islands (Fig. 3e).

**Magnetic properties of hydroxofluorographenes.** To exclude the effect of tetramethylammonium hydroxide and possible trace impurities contained in this precursor on magnetic behaviour of the system, we measured the temperature evolution of mass magnetic susceptibility ($\chi_{mass}$), confirming its solely diamagnetic character within the whole temperature range (Supplementary Fig. 5). In the case of other precursor, GF, the temperature profile of $\chi_{mass}$ can be fitted employing the Curie

law ($\chi_{mass} = C/T$, where $C$ is the Curie constant and $T$ is the temperature) and temperature-independent diamagnetic contribution (Supplementary Fig. 6). However, the paramagnetic response of GF is very weak due to minor defects in the GF structure; its magnetic behaviour is dominated by the diamagnetic term in accordance with the previous study[43]. Thus, both precursors used for synthesis of G(OH)F exhibit non-magnetic behaviour.

Importantly, the substitution of some of F atoms of GF by –OH groups produces a material with radically different magnetic properties. Observed magnetic features are not definitely driven by tiny impurities identified by ICP-MS as the sum of their $\chi_{mass}$, considering the magnetic moments of detected elements and recalculating their magnetic response to their weight in the sample and under an external magnetic field of 10 kOe, is four orders lower than the measured $\chi_{mass}$ of G(OH)F. Contrary to GF, the temperature dependence of $\chi_{mass}$ for G(OH)F is not described by the Curie or Curie–Weiss laws from 5 to 300 K, implying a magnetically ordered state up to room temperature. More strikingly, at room temperature, G(OH)F exhibits antiferromagnetic (AFM) behaviour as $\chi_{mass}$ (or $1/\chi_{mass}$) decreases (or increases) with lowering the temperature (Fig. 4a and Supplementary Fig. 7). Thus, at room temperature, G(OH)F shows AFM ordering, a behaviour never observed for any 2D organic material having only $s$ and/or $p$ electrons. At low temperatures, an abrupt increase in $\chi_{mass}$ is observed followed by a saturation trend, which is characteristic of ferromagnetic (FM) materials. The passage from the AFM to FM regime is also clearly evident from the $1/\chi_{mass}$ versus temperature curve with an inflection point at $\sim 62$ K (Supplementary Fig. 7). Thus, below $\sim 62$ K (the FM/AFM transition temperature,

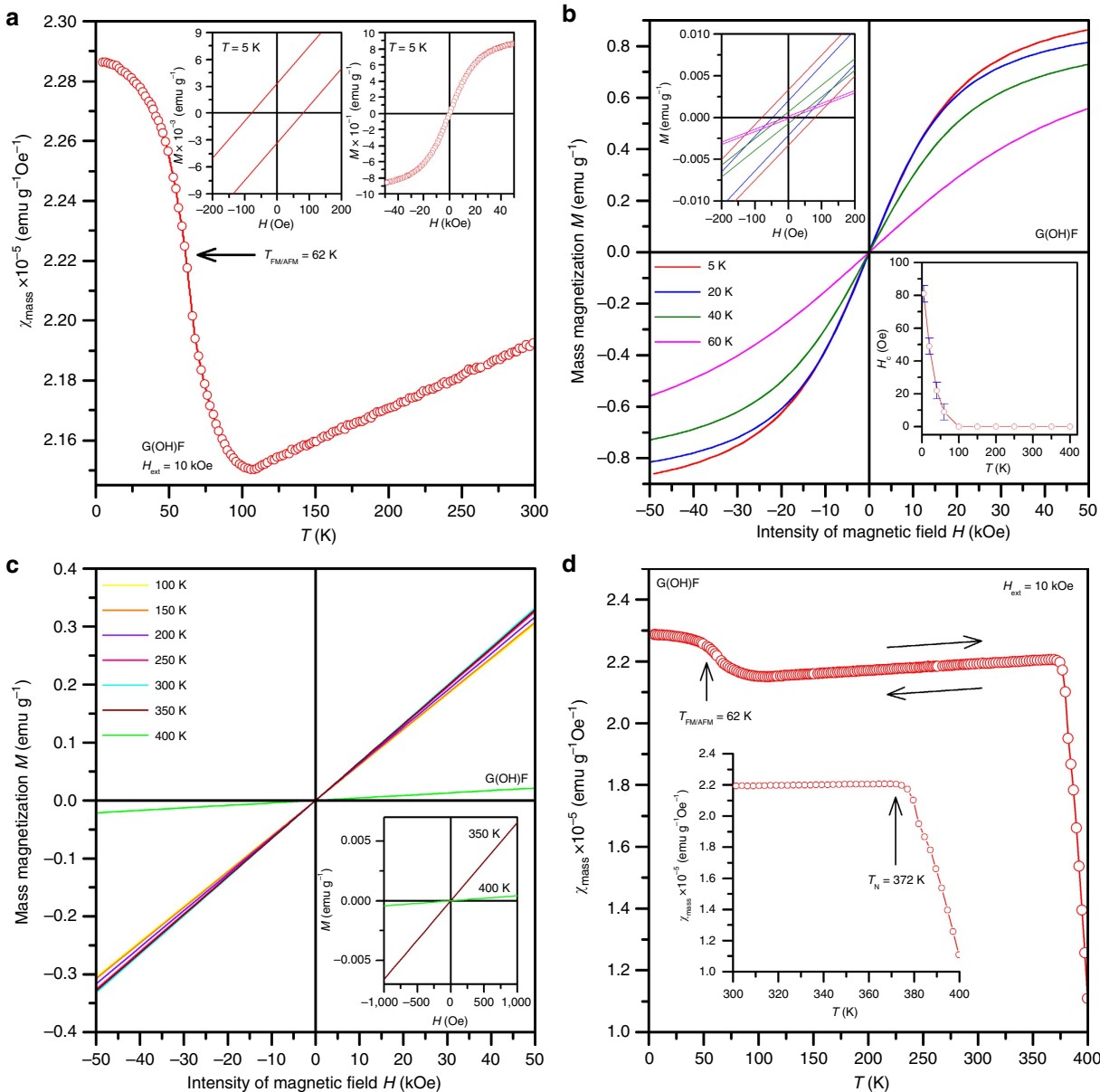

**Figure 4 | Magnetization measurements.** (**a**) Temperature evolution of the mass magnetic susceptibility ($\chi_{mass}$) of G(OH)F, measured under an external magnetic field ($H_{ext}$) of 10 kOe. The insets show the hysteresis loops of G(OH)F at 5 K, which indicate non-zero coercivity and a saturation magnetization of almost 1 emu g$^{-1}$. (**b**) Isothermal magnetization curves of G(OH)F at temperatures of 5–60 K. The insets show the profile of the hysteresis loops around the origin and the temperature dependence of coercivity ($H_C$). (**c**) Isothermal magnetization curves of G(OH)F, recorded from 100 to 400 K. The inset shows the profile of the isothermal magnetization curves at 350 and 400 K, demonstrating a dramatic decrease in the curve's gradient upon increasing temperature above 350 K; this implies a transition from an AFM state to a paramagnetic regime. (**d**) Temperature evolution of $\chi_{mass}$ of G(OH)F, measured under an external magnetic field of 10 kOe. The arrows show the reversibility of the $\chi_{mass}$ profile on warming the sample from 5 to 400 K and then cooling from 400 to 5 K. The inset depicts the behaviour of $\chi_{mass}$ between 300 and 400 K including its sudden drop above 370 K, which is indicative of a transition from an AFM state to the paramagnetic regime with a Néel transition temperature of about 372 K. Note: the paramagnetic signal from the non-interacting paramagnetic centres was subtracted from the $\chi_{mass}$ data.

$T_{FM/AFM}$), G(OH)F behaves in an FM manner. In other words, the FM state is a ground magnetic state for G(OH)F and the AFM regime can be regarded as thermally excited state. It is worth noting that similar FM-AFM transitions have been observed in various molecular radical-based systems[45]. The low-temperature FM state was further supported by G(OH)F isothermal magnetization curve, measured at 5 K: the curve exhibits hysteresis with a coercivity of ∼80 Oe (inset in Fig. 4a). Moreover, the magnetization of G(OH)F saturates above a value of 1 emu g$^{-1}$, placing G(OH)F among

the strong magnetic graphene-based systems including doped and functionalized graphenes and graphene derivatives. However, unlike other graphene-based materials, which also show FM features at low temperatures[13,28], G(OH)F synthesized as described above does not lose its magnetic ordering at room temperature; instead, it passes from an FM to AFM state as the temperature raises.

To address the evolution/sustainability of the magnetic properties of G(OH)F, we performed measurements of the material's hysteresis loops and $\chi_{mass}$ under heating from

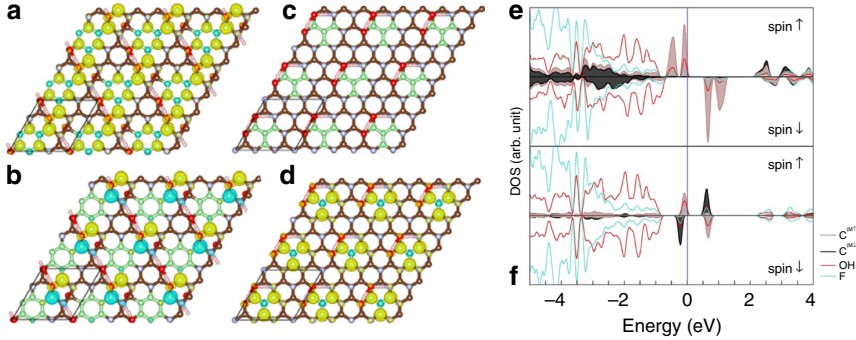

**Figure 5 | Spin densities and densities of states.** (**a,b**) The *m*-xylylene-like motif (green) of G(OH)F embedded in an $sp^3$ lattice with the corresponding FM (**a**) and AFM (**b**) phases, with up/down spin densities shown in yellow/blue. (**c,d**) $sp^3$-embedded trimethylenemethane-like motif with the corresponding FM spin density. Brown/green, blue, red and pink balls represent carbon, fluorine, oxygen and hydrogen atoms, respectively. (**e**) DOS of the GS FM phase. (**f**) DOS of the AFM phase. For (**e**) and (**f**), orbital contributions of individual atoms (labelled in the inset legend) are presented in both spectra. Electronic states of carbon atoms with up/down magnetic moments are plotted by brown/black line. Electronic states with the same spin direction can hybridize with each other. The hybridization involves the occupied and unoccupied spin-up states (**e**) and the occupied and unoccupied spin-up or spin-down states (**f**). Such kind of coupling not involving a finite density of states at $E_F$ is termed as the superexchange interaction. Both DOS spectra show a significant contribution of oxygen orbitals to the midgap states, which implies an important role of -OH group in the superexchange interactions.

5 to 400 K and cooling from 400 to 5 K (Fig. 4b–d and Supplementary Fig. 7). Between 5 and 60 K, the isothermal magnetization curves of G(OH)F reveal hysteresis with non-zero coercivity and remanent magnetization that both decrease with increasing temperature as would be expected for an FM ground state (Fig. 4b and its insets). Above 60 K, the hysteretic behaviour disappears and profiles of the isothermal magnetization curves are consistent with those expected for an AFM system (Fig. 4c). Importantly, above 350 K, the gradient of the magnetization versus field curve decreases sharply, implying a transition from an AFM to paramagnetic regime (inset in Fig. 4c,d), with the Néel temperature of $\sim 372$ K (see the sudden increase and/or drop in $1/\chi_{mass}$ and/or $\chi_{mass}$, respectively, above 370 K in Fig. 4d and Supplementary Fig. 7). Importantly, the profile of the $\chi_{mass}$ versus $T$ curve is reversible at temperatures of 5–400 K, meaning that mild thermal treatment does not cause any irreversible structural or magnetic changes in the material (arrows in Fig. 4d). In the paramagnetic region, the $\chi_{mass}$ profile fits well with the Curie–Weiss law ($\chi_{mass} = C/(T+\theta)$, where $\theta$ is the Weiss temperature) for AFM materials with $\theta \approx 186$ K. The difference between the values of $T_N$ and $\theta$ can be explained by different (not defects-like) origin of magnetic moments in the G(OH)F system and AFM state not being the ground magnetic state. More strikingly, the temperature evolution of coercivity and remanent magnetization in the FM state does not follow any formulas valid for *d*-electron driven magnetism (that is, Brillouin function, Bloch function and so on), further indicating a completely new source of magnetic moments—motifs—induced in the G(OH)F lattice (insets in Fig. 4b).

## Discussion

To explain the origin and sustainability of magnetism in G(OH)F systems and effect of chemical composition on magnetic features, we applied a high throughput theoretical screening of random structures of $C_{18}(OH)_yF_x$ (that is, with a large number (more than 1,400) of configurations, see Supplementary Table 2 listing abundances of FM and AFM ground states in % of $C_{18}(OH)_yF_x$), simulating thus reliably a random nature of the derivatization. We focused on identification of the nature of magnetic motifs driving room temperature magnetism, origin of FM-AFM transition, role of –OH groups in magnetic

communication of these motifs and effect of $C_{18}(OH)_yF_x$ (an F/OH ratio) on magnetic properties of the system.

Regarding the magnetism origin, we identified energetically stable diradical motifs, typically consisting of $sp^2$-conjugated islands embedded in an $sp^3$ matrix, which were responsible for the observed magnetic behaviour. A prototypical example is the *m*-xylylene motif, which comprises eight $sp^2$ carbon atoms and has an FM ground state (GS) with a spin moment $S=1$ and an FM-AFM spin-flip gap of 0.012 eV (Fig. 5a,b). Another important motif consists of four conjugated $sp^2$ carbon atoms in a triangular configuration that resembles trimethylenemethane (Fig. 5c,d). Structures containing the trimethylenemethane motif have a spin moment $S=1$ (FM GS) and an FM-AFM spin-flip gap of 0.012 eV. It is worth noting that both *m*-xylylene and trimethylenemethane are typical organic diradicals that are known sources of molecular magnetism[46]. Structural details of both prototypical diradical motifs are depicted in Supplementary Fig. 8. The presence of diradical motifs with FM GS and small FM-AFM spin-flip gap is consistent with the experimental FM-AFM transition observed for G(OH)F at $\sim 62$ K.

The electronic structure displayed as density of states (DOS) of G(OH)F with the *m*-xylylene-like motif both in the FM and AFM phase (Fig. 5e,f) provides insight into the nature of G(OH)F magnetism. The DOS in the vicinity of the Fermi level ($E_F$) is dominated by spin-polarized midgap states. Two midgap states per spin channel are visible in the FM phase (Fig. 5e), and only one channel is occupied. The midgap states reflect an imbalance of graphene bipartite lattice, $N = N_A - N_B$, where $N_A$ and $N_B$ are the number of atoms belonging to each sublattice. As the GS spin of imbalanced bipartite lattice is given by $2S = |N_A - N_B|$ (ref. 47), both diradical motifs have $N=2$ and, hence, $S=1$. The spin-polarized midgap states and bulk of positive (up) spin density reside within the diradical motif on the majority sublattice, while the electronic states of the minority sublattice form a band extending from 2.1 eV above $E_F$. Note that the electronic states with the same spin direction hybridize with each other.

The DOS plot indicates on the superexchange interactions[48] in maintaining the GS FM ordering within the diradical motifs. A substantial contribution of the oxygen *p* states of the bridging hydroxyl group (located between diradical motifs) to the midgap states (Fig. 5e) shows the important role of –OH groups in

stabilizing the FM GS through the coupling between magnetic diradical motifs. The importance of –OH functionalization in self-sustainability of magnetic ordering is further demonstrated by the fact that fully fluorinated analogue of the $m$-xylylene motif, that is, the system comprising $m$-xylylene-like $sp^2$ island surrounded by a fully fluorinated $sp^3$ network, is predicted to have a paramagnetic GS with an AFM-FM spin-flip energy of 0.001 eV (ref. 43). Introduction of a single –OH group to the supercell only slightly enhances the stability of the $m$-xylylene motif's FM state, by about $-0.03$ eV with respect to its fully fluorinated counterpart. A more significant stabilization of the FM GS was observed when two hydrogen-bonded –OH groups were introduced in close proximity to the $sp^2$ island; this increased the stability of the FM state relative to the AFM state by 0.08 eV. In addition, a strong overlap of the –OH and –F orbitals in a broad binding energy range starting from –18.5 eV highlights the importance of –OH groups preventing migration of fluorine atoms over the graphene surface and suppressing formation of non-magnetic islands, which are formed in partially fluorinated graphenes[43,49,50].

The DOS spectra of the AFM phase (Fig. 5f) are virtually identical as those for FM one up to the valence band maximum. The important differences appear in the occupation of the midgap states. The occupied (unoccupied) spin-up (spin-down) states are localized on the C atoms carrying positive spin density, while the opposite occupation resides on the C atoms with negative spin density. As hydroxyl group substantially contributes to the midgap states, we conclude that the AFM superexchange is driven by the magnetic coupling[48,51] in which –OH group plays the significant role. Needless to say that the orientation of the hydroxyl group does not affect the FM-AFM transition observed at $\sim 62$ K as both *ab initio* molecular dynamic simulations and arbitrary rotation of the –OH groups did not induce the phase change. Thus, the experimentally observed FM-AFM transition, inherent to organic-based materials with radical motifs[42], is in good agreement with a low-energy gain of superexchange interaction of the FM state, which is inversely proportional[52] to the energy difference between hybridized states ($\sim 2.6$ eV).

It is worth to mentioning that magnetism of graphene-based materials is intimately related to the appearance of midgap states in their electronic structure[28,29]. The origin of these midgap states is usually ascribed to defects or edges in the structure of graphene-based materials. Here we first identify the new principal source of magnetism in graphene derivatives based on creation of diradical domains through an appropriate $sp^3$ functionalization. Such diradical motifs embedded in an $sp^3$ lattice are thus clearly necessary but not sufficient for maintenance of the magnetic regime, which is further stabilized by the presence of neighbouring –OH groups. They show multiple roles in the system contributing to formation and stabilization of diradical motifs, superexchange interaction among them and suppression of adatoms lateral diffusion.

To support the principle role of –OH groups on magnetism, we further performed a high-temperature treatment of the G(OH)F sample. If it is exposed to a temperature above 200 °C, the room temperature magnetism is lost (after thermal treatment, G(OH)F behaves in a diamagnetic manner; Supplementary Fig. 9) as –OH groups start to leave the structure of G(OH)F (see, for comparison, Supplementary Fig. 3b). It confirms that –OH groups have an essential role on establishing and maintaining magnetically ordered state up to room temperature. In addition, defects such as vacancies, voids and edges cannot be considered as key factors capable to imprint the observed FM and AFM behaviour at low and room temperature, respectively.

Furthermore, we modelled the effect of the partial 3D layering of G(OH)F sheets on the magnetic features of the G(OH)F system. Here the individual layers in bulk G(OH)F samples were supposed to be bound by both dispersive forces and hydrogen-bonding between –OH groups. The theoretical calculations show that the magnetic properties of bulk G(OH)F are identical to those of the monolayered material (Supplementary Fig. 10). This indicates that magnetism of G(OH)F originates from the individual sheets and is not a consequence of their stacking. In other words, the potential contribution of 3D layering to the observed magnetism can be clearly ruled out.

To address the effect of chemical composition (OH and F contents) on magnetic properties of hydroxofluorographenes, we then constructed the magnetization map based on density functional theory (DFT) calculations for the $C_{18}(OH)_yF_x$ basic studied cell (Fig. 6). The magnetization map shows the propensity for the formation of FM ground states as a function of the F and OH content of the material (as also listed in Supplementary Table 2). Importantly, there are regions corresponding to the magnetically ordered FM ground states ranging from green to magnetically strongest red parts depending on the particular composition. Nevertheless, the blue regions corresponding to non-magnetic states are also present in a significant portion—the fact evidencing for a principal effect of chemical composition on magnetic features of G(OH)F system. In particular, two magnetically interesting islands appear, that is, upper island corresponding to G(OH)F systems with relatively high $sp^3$ functionalization (above $\sim 50\%$) and higher OH and F contents and lower island with considerably lower degree of $sp^3$ functionalization, centred at the $C_{18}F_4$ stoichiometry, surprisingly with very low OH content. Importantly, the stoichiometry of the G(OH)F sample, thoroughly discussed above, lies within the region of the upper island (sample denoted as No. 1 in the magnetization map in Fig. 6), where magnetism originates from the presence of $sp^2$-conjugated diradical motifs embedded in an $sp^3$ matrix. Computational predictions are thus consistent with available experimental data. In this region, the $sp^3/(sp^2 + sp^3)$ ratio is above the site percolation limit of honeycomb lattice (0.697)[52], which indicates that the number of $sp^3$ atoms is sufficient (above 5.5 for the supercell containing 18 carbon atoms) to cage the $sp^2$ islands.

To confirm the presence of diradical motifs experimentally, we prepared and measured the electron paramagnetic resonance (EPR) spectra (Supplementary Fig. 11) for the $C_{18}F_{11.5}$ sample whose $sp^3/(sp^2 + sp^3)$ ratio was above the percolation limit (0.735), favouring the emergence of diradical motifs according to our theory. This sample exhibited a similar level of $sp^3$ functionalization like G(OH)F sample, which is, however, antiferromagnetic ($S = 0$) and thus EPR silent. The EPR resonances recorded at two different temperatures clearly indicate the presence of radical species. In particular, overlapping signals originating from uncoupled $S = 1/2$ centres (strong resonance signals around $g \approx 2$) together with spin-coupled $S = 1/2$ systems were observed, leading to the formation of triplet species ($S = 1$). The EPR signatures of $S = 1$ systems exhibited small $E/D$ ratios (about 0.1) and became clearly visible upon lowering the temperature. The axial zero-field-splitting parameter ($D$) for the $S = 1$ system was found to be equal to $\sim 790$ MHz, which corresponds to an average distance of $\sim 4.6$ Å between the interacting $S = 1/2$ spins (point-dipole approach). Note that the distance between the two interacting $S = 1/2$ spins corresponds well with the distance between the methylene groups in the $m$-xylylene motif ($\sim 4.887$ Å). The EPR data thus unambiguously confirmed the presence

of diradical motifs in fluorographene systems of sufficiently high $sp^3$ content.

To validate the theoretical magnetization map, we further prepared four additional samples (samples denoted as No. 2–No. 5 in Fig. 6) exploiting various reaction conditions and –OH sources (for details, see 'Methods' section below) and determined their chemical composition by XPS (Supplementary Figs 12 and 13). Here it is worth noting that the experimental

scope for preparing the strongest magnetic systems will be limited by thermodynamics (Supplementary Fig. 14) and ability to prepare hydroxofluorographene systems with suitable and stable distributions of –OH and –F moieties (in this context, the facile migration of fluorines in lightly fluorinated graphenes may be problematic[43]). Nevertheless, the magnetic properties of all four additional derivatives with chemical compositions of $C_{18}(OH)_{1.5}F_6$, $C_{18}(OH)_2F_3$, $C_{18}(OH)_{2.6}F_{4.7}$ and $C_{18}(OH)_{2.4}F_7$

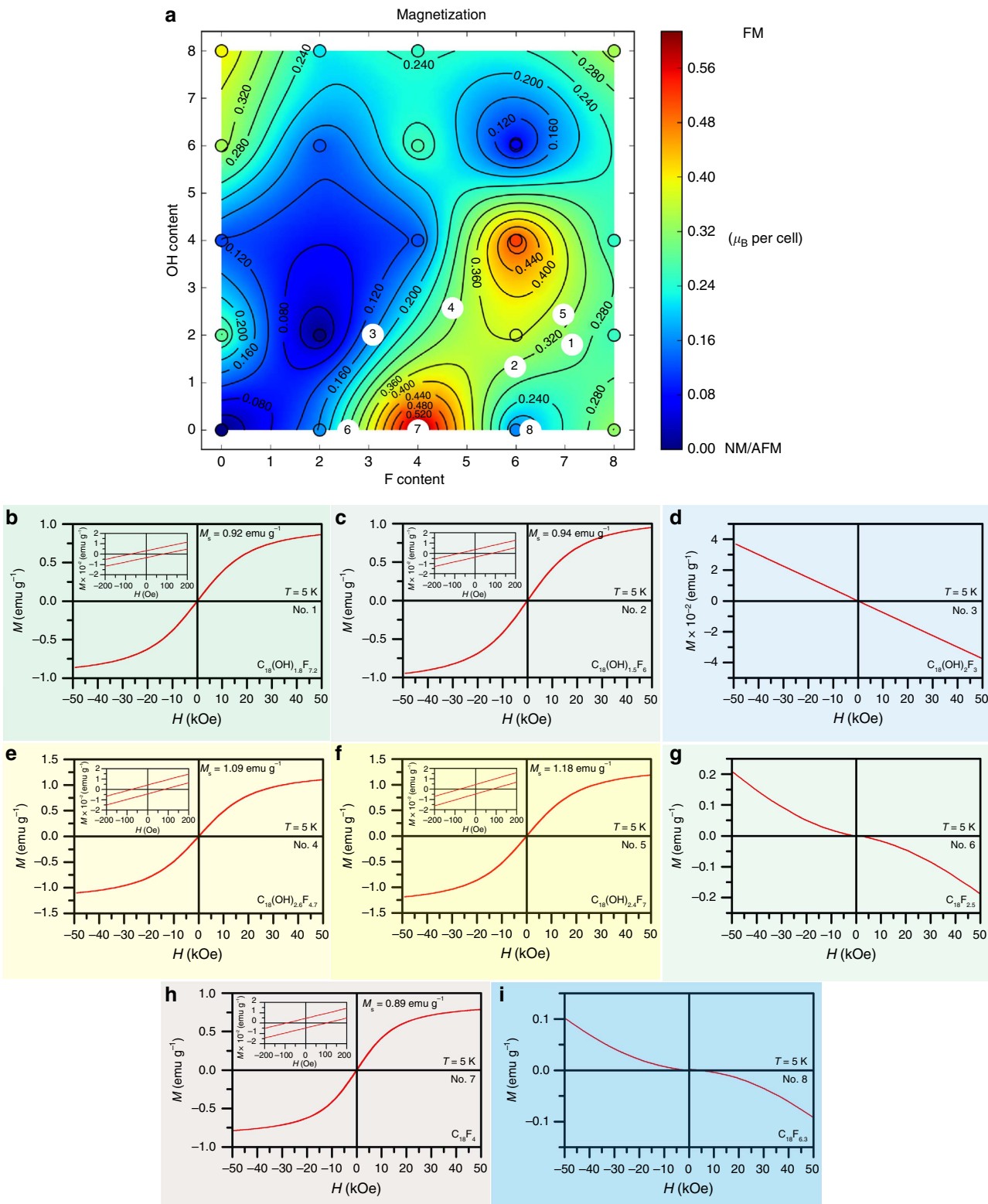

fit to the magnetization map with an excellent correlation and confirm the profound influence of the $C_{18}(OH)_yF_x$ stoichiometry (Fig. 6 and Supplementary Figs 12 and 13). In particular, $C_{18}(OH)_2F_3$ is a diamagnet, in full accordance with the map (it lies in one of the blue regions in Fig. 6). Conversely, $C_{18}(OH)_{1.5}F_6$, $C_{18}(OH)_{2.6}F_{4.7}$ and $C_{18}(OH)_{2.4}F_7$ are magnetically sustainable 2D materials with an FM GS (all lying in green-yellow region) and very similar transition temperature to the AFM state at $\sim 62$ K. The identical transition temperature, thus independent on chemical composition, is viewed as another proof of the same source of magnetism based on diradical motifs. The slight changes in saturation magnetization (from $\sim 0.9$ to $\sim 1.2$ emu g$^{-1}$; Fig. 6) stem from different strength of superexchange interactions associated to the content of –OH groups.

To further validate our theory, provide additional support for the essential role of –OH groups in the magnetism of hydroxofluorographenes, and explore the potentially different origins of magnetism in fluorographene and hydroxofluorographene systems, we also synthesized three partially fluorinated fluorographenes without –OH groups by simple thermal defluorination of the exfoliated $C_1F_1$ sample (for details on synthesis, see the 'Methods' section). The stoichiometries of these new systems are $C_{18}F_{2.5}$, $C_{18}F_4$ and $C_{18}F_{6.3}$ and the samples are denoted as No. 6, 7 and 8, respectively, in the magnetization map in Fig. 6 (for detailed XPS and magnetic characterization, see Supplementary Figs 15, 16 and 17). In full agreement with the magnetization map derived from our computational studies (Fig. 6), $C_{18}F_{2.5}$ is diamagnetic but contains some paramagnetic centres in perfect correspondence with the work by Nair et al.[43] reporting such behaviour for fluorographenes with lower fluorine coverages. Importantly, beyond the study by Nair et al.[43], the $C_{18}F_4$ exhibits even an FM GS as predicted by our magnetic model and experimentally confirmed (Fig. 6). However, it undergoes a transition to a paramagnetic state with the Curie temperature of 22 K. The different nature of transition temperature and inability to maintain magnetic ordering up to room temperature imply the different origin of magnetism in fluorographenes compared with hydroxofluorographenes. In particular, the structure of $C_{18}F_4$ can be considered as $sp^3$ structural defect in the $sp^2$ graphene lattice (Supplementary Fig. 18). This is further confirmed experimentally by detailed magnetic analysis of $C_{18}F_4$ (Supplementary Fig. 17). In particular, the temperature evolution of coercivity and saturation magnetization of $C_{18}F_4$ agrees very closely with predictions based on theoretical expressions for magnetism due to $d$-electrons (that is, models based on Brillouin and Bloch functions), implying that its magnetism is due to localized defect-induced magnetic moments[53,54]. These results starkly contrast with our observations for G(OH)F systems that exhibit room temperature magnetic ordering, whose experimental saturation

magnetization and coercivity data cannot be fitted using models based on expressions for $d$-electron magnetism. Finally, the $C_{18}F_{6.3}$ sample is again diamagnetic with some paramagnetic centres in agreement with the theoretical magnetization map (Fig. 6). Here the magnetic ordering is lost due to a lack of conduction electrons related to the increased degree of functionalization.

The comparison of magnetic behaviour of GF and G(OH)F systems demonstrates the differences between the 'defect-induced magnetism' observed for GF systems with lower levels of $sp^3$ functionalization and the 'diradical motif-based magnetism' observed in the highly functionalized G(OH)F system. At the same time, these data show that our theoretical model (that is, the magnetization map presented in Fig. 6) is robust, universal and capable of explaining the behaviour of both system types with different origins of magnetism.

In summary, we report a new class of graphene derivatives, which behave as antiferromagnetic materials at room temperature, representing examples of $sp$-based systems with room temperature magnetism. They become ferromagnets as the temperature is lowered showing remarkable magnetization. These organic magnets are prepared via simple, scalable and controllable reactions of fluorographene with suitable –OH-containing organic precursors. An interplay between thermodynamically preferred defluorination and nucleophilic substitution affects the products' final stoichiometry and thus their magnetic features. The magnetism in hydroxofluorographenes with an appropriate stoichiometry stems from the presence of diradical motifs coupled via superexchange interactions and stabilized by –OH groups, which also mediate the coupling. The newly constructed theoretical model addresses the effect of system stoichiometry on magnetic features in an excellent agreement with experimental data. More importantly, this robust model has a universal character covering the aspects of the 'defect-induced magnetism' and 'diradical motif-triggered magnetism' appearing in the field of graphene magnetism depending on degree of $sp^3$ functionalization.

We believe that this work would open the doors for preparing a wider family of graphene-based 2D room temperature magnets whose magnetic properties can be tuned by controlling the $sp^3$ functionalization. Definitely, the theoretical model based on diradical motifs communicating through superexchange interactions should be further extended also for other graphene-based systems. The developed room temperature carbon magnets also offer a huge space for testing in potential applications in various fields including, for example, spintronics and magnetically separable nanocarriers.

## Methods

**Chemicals.** Fluorinated graphite ($C_1F_1$) and tetramethylammonium hydroxide (25% w/w aqueous solution) were purchased from Sigma-Aldrich. Partly fluorinated graphites ($C_1F_{0.55}$ and $C_1F_{0.8}$) were purchased from Alfa Aesar.

**Figure 6 | Magnetic properties as a function of $C_{18}(OH)_yF_x$ stoichiometry.** (**a**) The mean magnetization map indicates which $C_{18}(OH)_yF_x$ ($x = 0–8$, $y = 0–8$) stoichiometries are likely to exist in ferromagnetic and non-magnetic ground states. Both –F and –OH groups are assumed to be randomly distributed across the sample (as suggested by STEM–HAADF elemental mapping, cf. Fig. 3) at zero temperature and the possibility of kinetically controlled –F/–OH migration is not considered. The white circles indicate experimentally studied samples: 1—$C_{18}(OH)_{1.8}F_{7.2}$ (ferromagnetic in the ground state), 2—$C_{18}(OH)_{1.5}F_6$ (ferromagnetic in the ground state), 3—$C_{18}(OH)_2F_3$ (diamagnetic in the ground state), 4—$C_{18}(OH)_{2.6}F_{4.7}$ (ferromagnetic in the ground state), 5—$C_{18}(OH)_{2.4}F_7$ (ferromagnetic in the ground state), 6—$C_{18}F_{2.5}$ (diamagnetic in the ground state), 7—$C_{18}F_4$ (ferromagnetic in the ground state) and 8—$C_{18}F_{6.3}$ (diamagnetic in the ground state). (**b**) The isothermal magnetization ($M$) curve of $C_{18}(OH)_{1.8}F_{7.2}$ (Sample No. 1) as a function of an external magnetic field ($H$), recorded at a temperature of 5 K. (**c**) $M$ versus $H$ curve of $C_{18}(OH)_{1.5}F_6$ (Sample No. 2), recorded at a temperature of 5 K. (**d**) $M$ versus $H$ curve of $C_{18}(OH)_2F_3$ (Sample No. 3), recorded at a temperature of 5 K. (**e**) $M$ versus $H$ curve of $C_{18}(OH)_{2.6}F_{4.7}$ (Sample No. 4), recorded at a temperature of 5 K. (**f**) $M$ versus $H$ curve of $C_{18}(OH)_{2.4}F_7$ (Sample No. 5), recorded at a temperature of 5 K. (**g**) $M$ versus $H$ curve of $C_{18}F_{2.5}$ (Sample No. 6), recorded at a temperature of 5 K. (**h**) $M$ versus $H$ curve of $C_{18}F_4$ (Sample No. 7), recorded at a temperature of 5 K. (**i**) $M$ versus $H$ curve of $C_{18}F_{6.3}$ (Sample No. 8), recorded at a temperature of 5 K. The insets in panel (**b,c,e,f** and **h**) show the behaviour of the respective hysteresis loops around the origin with the saturation magnetization ($M_S$) indicated.

DMF (p.a. grade) was obtained from PENTA, Czech Republic, and used without further purification.

**Detailed synthesis of samples.** To synthesize a few-layered fluorographene (GF) dispersion in DMF, fluorinated graphite (C:F, 1:1) (250 mg, Aldrich, grey powder) was suspended in 50 ml DMF. The mixture was sonicated for 2 h in an ultrasound bath and the temperature of the bath was kept below 30 °C. The suspension was then left to stand for 1 day to allow any undispersed material to settle. The clear, pale grey supernatant colloid was then collected and used in the preparation of hydroxofluorographene.

To synthesize hydroxofluorographene (G(OH)F, $C_{18}(OH)_{1.8}F_{7.2}$), 2 g of an aqueous tetramethylammonium hydroxide solution (25% w/w, Aldrich) was added to 50 ml of a fluorographene ($C_1F_1$) dispersion in DMF (prepared as described above) with a concentration of 0.8 mg ml$^{-1}$. The base initially precipitates in DMF but the precipitate gradually dissolves upon stirring. After 3 days' stirring at room temperature in a sealed vessel, the suspension was centrifuged at 5,000 r.p.m. for 10 min. The centrifuged solid was washed several times with water until its pH became neutral and then re-suspended in 6–8 ml water by sonication in an ultrasound bath (45 min). This dispersion was left to stand for 1 day to allow undispersed particles to settle and then centrifuged at 1,000 r.p.m. for 20 min to yield a clear black colloid containing aqueous dispersed hydroxofluorographene layers. Completely the same synthetic procedure was also applied to prepare hydroxofluorographenes ($C_{18}(OH)_2F_3$ and $C_{18}(OH)_{1.5}F_6$, respectively) from partly fluorinated graphites ($C_1F_{0.55}$ and $C_1F_{0.8}$, respectively).

Among several pathways tested for obtaining different hydroxylated graphene fluoride derivatives, two such reactions are described, as more appropriate for the scope of the present work. To synthesize $C_{18}(OH)_{2.6}F_{4.7}$, 50 mg of graphite fluoride was dispersed in 2 ml DMF and 0.7 ml $H_2O_2$ (30%, Aldrich) was added. The $C_{18}(OH)_{2.4}F_7$ sample was prepared by adding 0.7 ml tert-butyl hydroperoxide solution in decane (5.0–6.0 M, Aldrich) and 50 mg potassium tert-butoxide (Aldrich) in a 50 mg graphite fluoride dispersion in 2 ml DMF. Both mixtures were heated under stirring at 90 °C for 3 days. Work up of products was performed similarly with the previous ones.

Partially fluorinated fluorographenes, that is, with a composition of $C_{18}F_{2.5}$, $C_{18}F_4$ and $C_{18}F_{6.3}$, and the sample with the composition of $C_{18}F_{11.5}$ used for the EPR measurements, were obtained by thermal treatment of commercial fluorinated graphite under inert atmosphere at 800 °C for 4 h, at 625 °C for 4 h, at 550 °C for 5 h and at 550 °C for 1 h 52 min, respectively. The thermal treatments were performed in an open α-Al$_2$O$_3$ crucible using a Netzsch STA 449C Jupiter instrument. A temperature programme with a heating rate of 5 °C min$^{-1}$ (10 °C min$^{-1}$ for $C_{18}F_{11.5}$) from 40 °C to the final temperature was used. According to XPS analysis, the samples presented also small amounts of nitrogen and oxygen (atomic content, $C_{18}F_{2.5}$: N 2.0%, O 2.1%; $C_{18}F_4$: N 1.0%, O 1.4%; $C_{18}F_{6.3}$: O 0.6%; $C_{18}F_{11.5}$: O 3.1%).

**Thermal annealing of G(OH)F.** G(OH)F was thermally annealed by heating under inert atmosphere from room temperature to 220 °C at a rate of 5 °C min$^{-1}$, with a subsequent 3 h isotherm at 220 °C.

**Characterization techniques.** The exact composition of the precursors ($C_1F_{0.55}$, $C_1F_{0.8}$ and $C_1F_1$) and derived G(OH)F and partially fluorinated graphene (that is, $C_{18}F_x$, $x = 2.5, 4, 6.3$ and 11.5) samples was determined by XPS carried out with a PHI VersaProbe II (Physical Electronics) spectrometer using an Al K$_\alpha$ source (15 kV, 50 W). The obtained data were evaluated with the MultiPak (Ulvac-PHI, Inc.) software package. The detection of residual metal content in the G(OH)F sample was performed by ICP-MS. The exact amount of the G(OH)F sample (10 mg) was immersed in a concentrated nitric acid (≥99.999% trace metals basis) and heated for 2 h at 100 °C. Afterwards, the mixture was transferred into 10 ml volumetric flask, diluted with water and the undissolved graphene was caught by a 200 nm Millipore filter. The obtained concentration of metals in the solution was recalculated to the amount of the tested sample (analogically, diluted nitric acid was used as a blank). FT-IR spectra were recorded using an iS5 FT-IR spectrometer (Thermo Nicolet) with a Smart Orbit ZnSe ATR technique (650–4,000 cm$^{-1}$). Raman spectra were recorded on a DXR Raman microscope using the 532 nm excitation line of a diode laser. HRTEM images were obtained using an FEI TITAN 60–300 HRTEM microscope with an X-FEG type emission gun, operating at 300 kV. STEM–HAADF analyses for EDX mapping of elemental distributions on the G(OH)F sheets were performed with an FEI TITAN 60–300 HRTEM microscope operating at 80 kV. For HRTEM, STEM–HAADF and EDX experiments, an aqueous solution of G(OH)F with a concentration of 0.1 mg ml$^{-1}$ was redispersed by ultrasonication for 5 min. A drop of the sonicated sample was then deposited on a carbon-coated copper grid and slowly dried at laboratory temperature for 24 h to reduce its content of adsorbed water. Atomic force microscopy images and appropriate height profiles were recorded in semi-contact mode (HA-NC tips, mica substrate) on an NTEGRA Aura instrument. Thermo-gravimetric analysis and evolved gas analysis were performed on a STA449 C Jupiter-Netzsch instrument with a heating rate of 1 °C min$^{-1}$. The masses of released gases in the range of 12–60 m/z were determined with a QMS 403 Aolos mass spectrometer (Netzsch), starting at 100 °C to avoid overloading the

spectrometer with adsorbed water. A superconducting quantum interference device magnetometer (MPMS XL-7 type, Quantum Design, USA) was employed for the magnetization measurements. The temperature dependence of the magnetization of tetramethylammonium hydroxide, $C_1F_1$ precursor and seven final products (that is, $C_{18}(OH)_2F_3$, $C_{18}(OH)_{1.5}F_6$, $C_{18}(OH)_{2.6}F_{4.7}$, $C_{18}(OH)_{2.4}F_7$, $C_{18}F_{2.5}$, $C_{18}F_4$ and $C_{18}F_{6.3}$) was recorded in the sweep mode over the temperature interval from 5 to 300 K under an external magnetic field of 10 kOe; the temperature evolution of magnetization of the $C_{18}(OH)_{1.8}F_{7.2}$ sample was monitored in the sweep mode over the temperature interval from 5 to 400 K and back under an external magnetic field of 10 kOe. The hysteresis loops of the $C_{18}(OH)_{1.8}F_{7.2}$ and $C_{18}F_4$ samples were measured at a series of temperatures in the interval from 5 to 400 K and from 5 to 300 K, respectively, in external magnetic fields ranging from −50 to +50 kOe. The hysteresis loops of the $C_{18}(OH)_2F_3$, $C_{18}(OH)_{1.5}F_6$, $C_{18}(OH)_{2.6}F_{4.7}$, $C_{18}(OH)_{2.4}F_7$, $C_{18}F_{2.5}$ and $C_{18}F_{6.3}$ samples were measured at a temperature of 5 K and in external magnetic fields from −50 to +50 kOe. The magnetization values were corrected assuming the response of the sample holder, sample capsule and respective Pascal constants. EPR spectra were recorded on a JEOL JES-X-320 operating at X-band frequency (∼9.15 GHz), equipped with a variable temperature control ES 13060DVT5 apparatus, and were performed on the powder $C_{18}F_{11.5}$ sample (∼2 mg loaded onto the EPR tube). The cavity Q quality factor was kept above 6,000, highly pure quartz tube was employed (Suprasil, Wilmad, <0.5 OD) to load the sample powders. The g value accuracy was obtained by comparing the resonance signals of the $C_{18}F_{11.5}$ sample with that of MnO standard (JEOL-internal standard). The experimental conditions for measuring EPR spectra were adjusted as follows: frequency = 9.15584 GHz, modulation frequency = 100 kHz, modulation amplitude = 0.8 mT, time constant = 30 ms, applied microwave power = 0.8 mW, sweep time = 480 s and phase = 0°s.

**Computational methods.** Atomistic calculations were performed using the spin-polarized DFT and projected augmented wave potentials representing atomic cores as implemented in the VASP package[55–57]. The PBE xc functional[58] was used with a plane wave cutoff of 500 eV. The Brillouin zone integrations were performed with $3 \times 2 \times 1$ (structure and cell optimization) and $10 \times 10 \times 1$ (final runs; magnetism) Γ point-centred k-point Monkhorst-Pack meshes[59] per $3 \times 3$ supercell. The electronic density of states was calculated using the tetrahedron method[60] employed denser k-point sampling $16 \times 16 \times 1$. Open circles in Fig. 6 label the explored 21 unique stoichiometries $C_{18}(OH)_yF_x$. For each stoichiometry at least 64 random structures were generated. The optimized structures were converged to energy of less than $10^{-4}$ eV, and a convergence criterion of $10^{-6}$ eV for each SCF cycle. The thermodynamic stability of all reported configurations was analysed in terms of the stabilization energy

$$E_{stab} = [E(model) - N_F{}^{\star}E(F) - N_{OH}{}^{\star}E(OH) - N_C{}^{\star}E(C)]/(N_F + N_{OH} + N_C),$$

where $E(model)$, $E(F)$ and $E(OH)$ denote the total energy of the supercell model, F atom, –OH group and C atom, respectively and $N_F$, $N_{OH}$ and $N_C$ denote the number of F atoms, –OH groups and C atoms, respectively. The spin coupling constant or spin-flip gap per supercell was calculated as $E_{SF} = E_{LS} - E_{HS}$, where $E_{LS}$ and $E_{HS}$ are the total low-spin and total high-spin energy for a given configuration, respectively. The mean magnetization was obtained as average over 32 low-energy samples from 64 per cell as a function of x and y in the formula of $C_{18}(OH)_yF_x$. The colour-coded maps were generated by linear fitting among valid data points represented by circles.

To address the role of 3D stacking on the magnetic properties of the G(OH)F system, calculations including vdW-correction DFT-D3 of Grimme[61] were carried out for a hypothetical bulk. We considered typical stacking patterns, namely AA and AB; the relaxed interlayer distance for the most energetically stable structure was 5.9 Å. The energetically most stable structure together with the corresponding density of states plot is presented in Supplementary Fig. 10.

**Data availability.** The data that support the findings of this study are available from the corresponding author upon request.

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

## Acknowledgements

The authors acknowledge the support from the Ministry of Education, Youth and Sports of the Czech Republic (LO1305) and the assistance provided by the Research Infrastructure NanoEnviCz, supported by the Ministry of Education, Youth and Sports of the Czech Republic under Project No. LM2015073. M.O. acknowledges the Neuron fund for supporting science and the Czech Science Foundation (P208/12/G016) and ERC grant (ERC-2015-CoG). The authors deeply thank Dr. Giorgio Zoppellaro (from the Regional Centre of Advanced Technologies and Materials, Faculty of Science, Palacky University in Olomouc, Czech Republic) for measuring and interpreting the EPR spectra. The authors also thank Dr. Joelle Hoggan and her team at Sees-editing Ltd., UK, for professional language corrections of the manuscript.

## Author contributions

J.T. performed magnetization data analysis and participated in writing the manuscript, K.H. prepared the samples and performed data analysis, A.B.B. and J.U. prepared the samples, A.B. prepared the samples and performed data analysis, P.B., M.D. and F.K. carried out theoretical computations and participated in writing the manuscript, V.R. performed Raman and FT-IR measurements, K.C. performed microscopic experiments, M.O. carried out theoretical computations and participated in writing the manuscript and R.Z. came with the idea of imprinting magnetism to a graphene-based system through functionalization, designed the experiments and participated in writing the manuscript.

## Additional information

**Competing financial interests:** The authors declare no competing financial interests.

