## [Peer Review File · Nature Communications]

Reviewers' comments:

Reviewer #1 (Remarks to the Author):

The authors report an investigation of the magnetic properties of fluorohydroxygraphene of different stoichiometries by a wide range of experimental techniques. Some of the stoichiometries are found to exhibit an anomalous temperature dependence of the magnetic susceptibility with a hysteresis in the magnetization curves observed at low temperatures. This behavior is interpreted as the onset of antiferromagnetic ordering with Neel temperatures above the room temperature, followed by the transition to ferromagnetic ordering at low temperatures. First-principles calculations are presented to support the hypothesis that such a behaviour can be explained from the point of view of the presence of biradical motifs.

The field of carbon-based magnetism has been attracting considerable attention because of potential technological applications of such materials. However, the exact physical mechanisms underlying the observed unusual magnetic properties are still under discussion. This is not surprising considering the complex structure and intrinsically disordered nature of the discussed materials. The present manuscript is a very thorough investigation, certainly among the best in this field. However, I feel like certain claims are exaggerated and several issues need to be resolved. I cannot recommend publishing this work in its present form, but I would be glad to consider a revised version.

- The interplay between ferro- and antiferromagnetic interactions in graphene-based materials is not surprising as such, but I'm not sure that the observed complex behavior of magnetic susceptibility can be unambiguously interpreted as an onset of antiferromagnetic state with high Neel temperatures followed by a transition to ferromagnetic state at low temperatures. Perhaps, investigating magnetic susceptibility above the Neel temperature can provide some useful information. Does it follow the $\chi \sim 1/(T + \theta)$ dependence expected for antiferromagnets? Is the value of θ consistent with the Neel temperature discussed in the paper?

- One important question concerns the reproducibility of results, which is a necessary attribute of any scientific work. The authors show that the magnetic properties of fluorohydroxygraphenes are very sensitive to stoichiometry. But do susceptibility curves for two different samples of the same stoichiometry agree with each other?

- The theoretical part pointing at the crucial role of biradical species is thoroughly done. However, an explicit experimental evidence of the presence of such moieties needs to be presented. Otherwise, one can speak only about a hypothesis.

- I'm not sure about the meaning of word "self-sustainable".

Reviewer #2 (Remarks to the Author):

This is highly important paper. It shows conceptually new magnetism of materials, not originating from d-elements but instead based on sp³ graphene derivatives. This work is seminal and opens door for whole field.

The manuscript is carefully drafted, contains detailed experimental analysis which is in very good agreement with theory. I do not see major issues here. I support publication of this manuscript after following issues are addressed.

1) Theory is based on unit cell of definite size. Is it possible to take in account i.e. 3 different derivatizations at different points (that is, the material with the same summary formula but different distribution of functional groups) and to calculate "average" magnetic properties of the material? This would reflect better random nature of the derivatization of real materials.

2) It is known that graphene materials can contain metallic impurities in some cases. It is important to mention this, cite appropriate literature and guide readers how to distinguish between d-element impurities magnetism and sp³ magnetism to have really strong impact on the field.

Reviewer #3 (Remarks to the Author):

In this paper, authors reported an easy approach to synthesize stable graphene-based materials, namely hydroxyfluorographenes, G(OH)F, with self-sustainable magnetism up to room temperature. This series of G(OH)F were experimentally prepared from fluorographene and N(CH₃)₄OH. The authors tested different ratio of F and OH to detect the magnetism and investigated the mechanism of this self-sustainable magnetism both experimentally and theoretically. In general, this is a well conceived and carefully performed work, publication is recommended after the following issues are considered.

1. The authors stated "the first organic sp-based magnets with self-sustainable magnetism up to room temperature". Many strategies have been attempted to introduce the magnetism to graphene and its derivatives, such as surface modification, N/O/F absorption, Electric field engineering, Co atom adsorption. Therefore, the statement in lines 22-23, page 2, may be inappropriate.

2. It's not the first attempt to introduce the magnetism to graphene by surface modification. Thus, the authors should further present the importance of this work.

3. The computational models are not clear enough to distinguish the configurations. Maybe the authors can present the configurations in more details in the supporting information.

4. In the "Computational methods" (Line 490-492, page 22), some refs should be cited for the VASP code and other computational details (PBE functional, PAW potential, et al).

Authors' replies to the Reviewer's comments and questions

Reviewer #1

Reviewer's Comment: The authors report an investigation of the magnetic properties of fluorohydroxygraphene of different stoichiometries by a wide range of experimental techniques. Some of the stoichiometries are found to exhibit an anomalous temperature dependence of the magnetic susceptibility with a hysteresis in the magnetization curves observed at low temperatures. This behavior is interpreted as the onset of antiferromagnetic ordering with Neel temperatures above the room temperature, followed by the transition to ferromagnetic ordering at low temperatures. First-principles calculations are presented to support the hypothesis that such a behaviour can be explained from the point of view of the presence of biradical motifs.

The field of carbon-based magnetism has been attracting considerable attention because of potential technological applications of such materials. However, the exact physical mechanisms underlying the observed unusual magnetic properties are still under discussion.

This is not surprising considering the complex structure and intrinsically disordered nature of the discussed materials. **The present manuscript is a very thorough investigation, certainly among the best in this field. However, I feel like certain claims are exaggerated and several issues need to be resolved.** I cannot recommend publishing this work in its present form, but I would be glad to consider a revised version.

Reviewer's Comment 1: The interplay between ferro- and antiferromagnetic interactions in graphene-based materials is not surprising as such, but I'm not sure that the observed complex behavior of magnetic susceptibility can be unambiguously interpreted as an onset of antiferromagnetic state with high Neel temperatures followed by a transition to ferromagnetic state at low temperatures. Perhaps, investigating magnetic susceptibility above the Neel temperature can provide some useful information. Does it follow the $\chi \sim 1/(T + \theta)$ dependence expected for antiferromagnets? Is the value of θ consistent with the Neel temperature discussed in the paper?

Reply to the Reviewer's Comment 1: We fully agree with the Reviewer. If defects are introduced into the graphene lattice, ferromagnetic and/or antiferromagnetic ordering can be established depending on the nature and quantity of the defects present on the two graphene sublattices. This is consistent with the predictions of the Lieb's theorem (e.g., Yazyev, O. V. Rep. Prog. Phys. **73**, 056501 (2010), Lieb, E. H. Phys. Rev. Lett. **62**, 1201 (1989)). Ferromagnetic and antiferromagnetic ordering can coexist in graphene-based materials if the interactions between defect-induced magnetic moments are of the RKKY type, with the π -electrons of the graphene skeleton serving as mediators.

However, the origin of magnetism in hydroxofluorographene systems based on the presence of superexchange interacting diradical motifs is of distinct nature compared to “traditional d-element based magnetism”. This is reflected in temperature dependence of hysteresis parameters as thoroughly discussed on pages 13-14. When fitting the magnetic susceptibility data of the prototypical hydroxofluorographene system in the paramagnetic regime (i.e., above 372 K), the data follow the Curie-Weiss law as typical for antiferromagnetic materials. Particularly, the θ parameter was found to equal to + 186 K which is far from the experimentally observed Néel temperature ($T_N = 372$ K). This difference between the θ parameter and Néel temperature is not surprising taking into account the above-mentioned different origin of magnetism and the fact that the ferromagnetic regime is the ground magnetic state. The transition from the FM to the AFM state observed here at 62 K is related to a low energy gain for superexchange interactions in the FM state, which is inversely proportional to the energy difference between hybridized states (~ 2.6 eV), as already discussed in the manuscript (see Page 13). It is worth noting that similar FM-AFM transitions have been observed in various molecular radical-based systems (e.g., Cho, D. J. et al., *Phys. Chem. A* **118**, 5112 (2014)). It thus seems that such magnetic transitions are inherent to organic-based materials with radical motifs. In summary, because the AFM state is not the ground magnetic state for the studied hydroxofluorographenes, T_N does not match with θ . A discussion of FM-AFM transitions in molecular radical-based systems and the discrepancy between T_N and θ has been newly added to the revised manuscript (see Page 10).

Reviewer’s Comment 2: One important question concerns the reproducibility of results, which is a necessary attribute of any scientific work. The authors show that the magnetic properties of fluorohydroxygraphenes are very sensitive to stoichiometry. But do susceptibility curves for two different samples of the same stoichiometry agree with each other?

Reply to the Reviewer’s Comment 2: As the Reviewer correctly states, the magnetic behavior of hydroxofluorographenes depends strongly on their stoichiometry, i.e., the F/OH ratio. In other words, the ground magnetic state of hydroxofluorographenes may be diamagnetic, paramagnetic, ferromagnetic and/or antiferromagnetic depending on the F/OH ratio, as clearly demonstrated by our theoretical and experimental results. All the experimental samples (8) analyzed in our study have been synthesized three or more times; the key sample, which has a stoichiometry of $C_{18}(OH)_{1.8}F_{7.2}$, has been synthesized five times. Within the experimental error of the magnetization measurements, all samples of the same stoichiometry yielded identical susceptibility measurements, confirming the reproducibility of these results.

Reviewer’s Comment 3: The theoretical part pointing at the crucial role of biradical species is thoroughly done. However, an explicit experimental evidence of the presence of such moieties needs to be presented. Otherwise, one can speak only about a hypothesis.

Reply to the Reviewer's Comment 3: Following the Reviewer's suggestion, we used EPR spectroscopy to confirm the presence of diradical motifs in the hydroxofluorographene samples. Because the EPR signal is silent for antiferromagnetic materials (which have zero net magnetic moment), we chose a partially defluorinated fluorographene sample ($C_{18}F_{11.5}$; see the newly added Supplementary Figure 11 – EPR spectra and XPS pattern) whose $sp^3/(sp^2 + sp^3)$ ratio (0.735) is above the percolation limit (0.697), favoring the emergence of diradical motifs according to our theory. The EPR resonances recorded at two different temperatures for the $C_{18}F_{11}$ sample clearly indicate the presence of radical species. In particular, there are overlapping signals originating from uncoupled $S = 1/2$ centers (the strong resonance signals around $g \approx 2$ in spectra (a) and (b)) together with spin-coupled $S = 1/2$ systems, leading to the formation of triplet species ($S = 1$). The EPR signatures of the $S = 1$ systems exhibit small E/D ratios (about 0.1) and became clearly visible upon lowering the temperature (see EPR spectrum b), although their presence could be detected even at higher temperatures, as shown by the x10 magnification of the low field region of spectrum (a). The axial zero-field-splitting parameter (D) for the $S = 1$ system was estimated to take a value of $|D| \approx 790$ MHz, which corresponds to an average distance between interacting $S = 1/2$ spins of ≈ 4.6 Å based on the point dipole approach. Note that distance between the two interacting $S = 1/2$ spins corresponds well with the distance between the methylene groups in the *m*-xylylene motif. The EPR data thus unambiguously confirmed the presence of diradical motifs in fluorographene systems of sufficiently high sp^3 content (see newly involved discussion on Page 15). Moreover, the presence of radical motifs in fluorinated graphenes has previously been observed by EPR, albeit without any deep discussion of their consequences in magnetic sustainability (e.g., Nair, R. R. et al., Nat. Phys. **8**, 199 (2012); Panich, A. M. et al., J. Phys. Chem. Solids **62**, 959 (2001)). Thus, both experiment and theory confirm that diradical motifs are the key source of ground state magnetism in hydroxofluorographenes, and that –OH groups are crucial mediators enabling their magnetism to be sustained up to room temperature.

Supplementary Figure 11. EPR measurements. X-band EPR spectra of the $C_{18}F_{11.5}$ sample recorded at (A) $T = 293$ K and (B) $T = 173$ K. The low-field region of the spectrum has been magnified ($\times 10$, upper trace) to more clearly show the presence of triplet species coexisting with the doublet species. The EPR spectra are averaged and accumulated composites of two scans. The inset shows the high-resolution C 1s XPS pattern provide quantification of the $sp^3/(sp^2 + sp^3)$ ratio (0.735) for the $C_{18}F_{11.5}$ sample.

Reviewer’s Comment 4: I’m not sure about the meaning of word "self-sustainable".

Reply to the Reviewer’s Comment 4: We fully agree with this comment and apologize for the use of unclear terminology. The adjective “self-sustainable” refers to the material’s ability to sustain magnetic ordering at elevated temperatures. In other words, it refers to an inherent property of the material whereby magnetic ordering is established spontaneously through the evolution of dipolar or exchange interactions. Thus, no external stimulus (such as an external magnetic field) is needed to orient the magnetic moments (or spins) along a particular axis. However, despite of many graphene based “self-sustainable” systems reported in the literature (with transition temperatures ranging from 1 to 100 K), this is for the first time when the magnetism is preserved up to room temperature. Thus, we decided to replace the confusing term “self-sustainable” with “room temperature” within the whole manuscript, thus stressing the novelty of the work.

Reviewer #2

Reviewer's Comment: This is highly important paper. It shows conceptually new magnetism of materials, not originating from d-elements but instead based on sp^3 graphene derivatives. This work is seminal and opens door for whole field.

The manuscript is carefully drafted, contains detailed experimental analysis which is in very good agreement with theory. I do not see major issues here. I support publication of this manuscript after following issues are addressed.

Reviewer's Point #1: Theory is based on unit cell of definite size. Is it possible to take in account i.e. 3 different derivatizations at different points (that is, the material with the same summary formula but different distribution of functional groups) and to calculate "average" magnetic properties of the material? This would reflect better random nature of the derivatization of real materials.

Reply to Reviewer's Point #1: We greatly appreciate this comment. We would like to stress that our theoretical models are based on a high throughput computational screen of randomly generated $C_{18}(OH)_x F_y$ structures with many (over 1400) different configurations. For each stoichiometry, at least 64 random structures (different distributions of functional groups) were generated. The mean magnetization as a function of the parameters x and y in the stoichiometric formula $C_{18}(OH)_y F_x$ was computed by averaging over the 32 lowest energy samples from 64 per cell. The color-coded maps were then generated by linear fitting among valid data points (represented by circles). This enabled us to model the random functionalization of the graphene sheets in our computational studies. The use of randomness in the computational studies is explained more carefully when introducing the computational model in the revised manuscript (see Page 11).

Reviewer's Point #2: It is known that graphene materials can contain metallic impurities in some cases. It is important to mention this, cite appropriate literature and guide readers how to distinguish between d-element impurities magnetism and sp^3 magnetism to have really strong impact on the field.

Reply to Reviewer's Point #2: We completely agree with the Reviewer. If metallic impurities are present (even at very low concentrations) they will always overshadow the magnetic response from graphene-based materials, leading to misinterpretation of results and inaccurate conclusions. Following the reviewer's recommendation, we have added new sentences to the Introduction of the manuscript stressing the importance of excluding d-block element impurities in graphene-based systems (with citations of appropriate works). In the Results section, we emphasize the point that a material exhibiting diradical motif-induced magnetism can be clearly distinguished from one exhibiting classical d-magnetism by an inability to correctly fit the temperature evolution of its magnetic parameters (i.e., magnetic susceptibility, coercivity, saturation magnetization) using equations and relationships derived for d- and f-orbital driven magnetism (see Pages 10-11 and Page 17 of the revised manuscript). Any

contribution of d-elements to the observed sp-magnetism is clearly excluded by detailed analysis of the ICP and magnetic data presented on Page 6 and in Supplementary Table 1.

Reviewer #3

Reviewer's Comment: In this paper, authors reported an easy approach to synthesize stable graphene-based materials, namely hydroxyofluorographenes, G(OH)F, with self-sustainable magnetism up to room temperature. This series of G(OH)F were experimentally prepared from fluorographene and N(CH₃)₄OH. The authors tested different ratio of F and OH to detect the magnetism and investigated the mechanism of this self-sustainable magnetism both experimentally and theoretically. **In general, this is a well conceived and carefully performed work, publication is recommended after the following issues are considered.**

Reviewer's Point #1: The authors stated “the first organic sp-based magnets with self-sustainable magnetism up to room temperature”. Many strategies have been attempted to introduce the magnetism to graphene and its derivatives, such as surface modification, N/O/F adsorption, Electric field engineering, Co atom adsorption. Therefore, the statement in lines 22-23, page 2, may be inappropriate.

Reply to Reviewer's Point #1: As stated in the Introduction, several methods for creating magnetically active graphene derivatives have been explored. We thank the Reviewer for specifying a few of them. The references related to electric field engineering and light element adsorption were involved in the revised version of the manuscript (please see Refs. 38-40, 42). Indeed, there are also the strategies based on adsorption of transition metal adatoms, which we newly cite in the revised version (please, see Ref. 41). However, all the strategies reported to date have been based on creating defects that introduce sp^3 states into the graphene lattice, which ultimately leads to the formation of spin polarized states at the Fermi level. These defect-induced paramagnetic centres can cooperate if suitable interaction media/pathways are available. In all cases reported to date, sp^2 conduction electrons were identified as the mediators of the interactions between paramagnetic centres, leading to the establishment of a magnetically ordered state (ferromagnetic and/or antiferromagnetic) *but only at low temperatures (generally below 100 K). Theoretical predictions and experimental observations suggest that such interactions are weak and readily disrupted by thermal fluctuations; as such, they cannot be used to sustain magnetic ordering at elevated temperatures up to room temperature.* Our work introduces a new source of magnetism based on diradical motifs created through a relatively high degree of sp^3 functionalization. The presence of -OH groups enables the formation of communication pathways between these paramagnetic centres based on superexchange interactions. These interactions appear to be stronger than those mediated by sp^2 conduction electrons and can thus stabilize magnetically ordered states at up to room temperature. Thus, we decided to replace the confusing term “self-sustainable magnetism” with

“room-temperature magnetism” within the whole manuscript, thus stressing the novelty of the work (please see also our Reply #4 to the Reviewer #1).

Reviewer’s Point #2: It’s not the first attempt to introduce the magnetism to graphene by surface modification. Thus, the authors should further present the importance of this work.

Reply to Reviewer’s Point #2: We fully agree with the Reviewer that this is not the first attempt to create magnetically active graphene derivatives by surface modification, but (as all of the reviewers acknowledged) it is the first report of a graphene derivative that retains its magnetism **at room temperature**, which open up a wide range of potential applications. *All previously reported magnetically active graphene derivatives exhibit magnetism at low temperatures but cannot sustain magnetic ordering up to room temperature.* This is due to their reliance on a different (defect-induced) origin of magnetism to that exhibited by hydroxofluorographenes (please see the previous response). Following the Reviewer’s recommendation, we have rephrased some sentences in the Abstract, Introduction, and Concluding sections of the manuscript to better highlight the significance/importance of the work, with particular emphasis on the point that *room temperature magnetism has never previously been observed in graphene-based materials.* Additionally, we have reinforced the point that this room temperature magnetism stems from the presence of diradical motifs coupled via superexchange interactions and stabilized by –OH groups, which also mediate the coupling. Moreover, the newly constructed theoretical model addresses the effect of system stoichiometry on magnetic features in an excellent agreement with experimental data. Following the recommendation of Reviewer #1, we newly added EPR data (see Supplementary Figure 12) confirming the presence of diradical motifs in a good agreement with a theory (please see also our Reply #3 to the Reviewer #1).

Reviewer’s Point #3: The computational models are not clear enough to distinguish the configurations. Maybe the authors can present the configurations in more details in the supporting information.

Reply to Reviewer’s Point #3: We thank the Reviewer for his/her comment. We added a new figure to the Supplementary Online Information file (please, see Supplementary Fig. 8) showing structural details of the prototypical motifs. The insets of this figure depict the structures of the two diradical motifs.

A *m*-xylylene in $C_xF_y(OH)_z$

B trimethylenemethane motif in $C_xF_y(OH)_z$

Supplementary Figure 8. Models of diradical motifs in the G(OH)F system. (A) *m*-xylylene motif in $C_xF_y(OH)_z$. (B) Trimethylenemethane motif in $C_xF_y(OH)_z$. The left and middle panels show the diradical motifs from above and below the graphene plane, while the right panels shows the details of the particular diradical motif. The insets in the right panels show the structures of the two diradical motifs. Fluorines are shown in yellow, hydrogens in white, oxygens in red, sp^3 carbons in blue, and sp^2 carbons in grey.

Reviewer's Point #4: In the "Computational methods" (Line 490-492, page 22), some refs should be cited for the VASP code and other computational details (PBE functional, PAW potential, et al).

Reply to Reviewer's Point #4: We thank the Reviewer for this comment. Citations of the appropriate works have been added to the Methods section. The new references are numbered from 55 to 61 in the list of references.

REVIEWERS' COMMENTS:

Reviewer #1 (Remarks to the Author):

The authors adequately addressed my comments and improved their manuscript. I feel free to recommend it for publication.

Reviewer #2 (Remarks to the Author):

The authors addressed well all my questions. This paper is of high importance and it will be highly followed and cited.

Minor note. Ref 23 is a bit outdated. Authors may consider to add newer reference to GO, i.e. nice review by Nishina's group in Appl. Mater. Today 2015, 1, 1.

Reviewer #3 (Remarks to the Author):

All the concerns in my previous report have been well addressed. Publication is recommended.

Authors' replies to the Reviewer's comments and questions (Manuscript ID: NCOMMS-16-23531A)

Reviewer #1

Reviewer's Comment: The authors adequately addressed my comments and improved their manuscript. I feel free to recommend it for publication.

Reply to the Reviewer's Comment: We thank the Reviewer very much for his/her positive feedback on the revision we did. We deeply appreciate the Reviewer's comments and suggestions that certainly improved the quality of the work.

Reviewer #2

Reviewer's Comment: The authors addressed well all my questions. This paper is of high importance and it will be highly followed and cited.

Minor note. Ref 23 is a bit outdated. Authors may consider to add newer reference to GO, i.e. nice review by Nishina's group in Appl. Mater. Today 2015, 1, 1.

Reply to the Reviewer's Comment: As recommended by the Reviewer, we replace the outdated Ref. 23 with the suggested review work. We thank the Reviewer for his/her positive feedback on the revision we did. We deeply appreciate the Reviewer's comments and suggestions that certainly improved the quality of the work.

Reviewer #3

Reviewer's Comment: All the concerns in my previous report have been well addressed. Publication is recommended.

Reply to the Reviewer's Comment: We thank the Reviewer very much for his/her positive feedback on the revision we did. We deeply appreciate the Reviewer's comments and suggestions that certainly improved the quality of the work.